# Optimisation of Sporosori Purification and Protein Extraction Techniques for the Biotrophic Protozoan Plant Pathogen *Spongospora subterranea*

**DOI:** 10.3390/molecules25143109

**Published:** 2020-07-08

**Authors:** Sadegh Balotf, Richard Wilson, Robert S. Tegg, David S. Nichols, Calum R. Wilson

**Affiliations:** 1Tasmanian Institute of Agriculture, New Town Research Laboratories, University of Tasmania, New Town, Tasmania 7008, Australia; sadegh.balotf@utas.edu.au (S.B.); robert.tegg@utas.edu.au (R.S.T.); 2Central Science Laboratory, University of Tasmania, Hobart, Tasmania 7001, Australia; d.nichols@utas.edu.au

**Keywords:** *Spongospora subterranea*, sporosori, density gradient centrifugation, Ludox^®^, proteomics, S-Trap

## Abstract

*Spongospora subterranea* is a soil-borne plant pathogen responsible for the economically significant root and powdery scab diseases of potato. However, the obligate biotrophic nature of *S. subterranea* has made the detailed study of the pathogen problematic. Here, we first compared the benefits of sporosori partial purification utilizing Ludox^®^ gradient centrifugation. We then undertook optimization efforts for protein isolation comparing the use of a urea buffer followed by single-pot solid-phase-enhanced sample preparation (SP3) and a sodium dodecyl sulphate (SDS) buffer followed by suspension-trapping (S-Trap). Label-free, quantitative proteomics was then used to evaluate the efficiency of the sporosori purification and the protein preparation methods. The purification protocol produced a highly purified suspension of *S. subterranea* sporosori without affecting the viability of the spores. The results indicated that the use of a combination of SDS and S-Trap for sample clean-up and digestion obtained a significantly higher number of identified proteins compared to using urea and SP3, with 218 and 652 proteins identified using the SP3 and S-Trap methods, respectively. The analysis of proteins by mass spectrometry showed that the number of identified proteins increased by approximately 40% after the purification of spores by Ludox^®^. These results suggested a potential use of the described spore purification and protein preparation methods for the proteomics study of obligate biotrophic pathogens such as *S. subterranea*.

## 1. Introduction

Plasmodiophorid organisms include several important plant pathogens that cause significant losses in a range of species either directly [1,2] or as vectors of plant viruses [3]. A common feature of plasmodiophorids is their environmental persistence through the production of long lived resting spores. Resting spore formation and sporosori complexity varies between the members of the plasmodiophorid genera [4]. *Plasmodiophora* produces individual non-aggregated resting spores, while *Polymyxa*, *Spongospora*, *Sorodiscus*, *Tetramyxa*, *Octomyxa*, *Woronina* and *Ligniera* all produce sporosori of variable shape and size [5,6,7,8,9]. Among all the known plasmodiophorids, *Spongospora* has the most complex sporosori with a sponge-like structure of variable size [10].

*S. subterranea* is the causal agent of root and powdery scab diseases of potato, of economic importance wherever potatoes are grown [11]. The pathogen can persist in infested soils for years, providing inoculum for subsequent potato crops and negating much of the benefits of crop rotation [12]. The pathogen produces large sporosori, comprising of an aggregation of hundreds to thousands of long-lived resting spores [13]. The resting spores within the sporosori have phases of both constitutive and exogenous (stimuli-responsive) dormancy [14]. The processes of infection and disease development in the *S. subterranea*–potato pathosystem have been well examined. However, the complex structure of the sporosori, the obligate biotrophic nature of *S. subterranea* and the shortcomings in current methodology for the separation of resting spores from the host tissues have hindered the application of new technologies for the detailed molecular analysis of these processes.

Density gradient centrifugation, developed by Brakke [15], has proven to be a feasible method for the isolation of particles of various structures and sizes. Harrison and Nixon [16] introduced density gradient centrifugation to soil-borne disease research by using a sucrose density gradient method for the purification of three soil-borne plant viruses. Castlebury et al. [17] developed a density gradient technique for the purification of *Plasmodiophora brassicae* resting spores from infected Chinese cabbage host root tissue. Resting spores were first extracted from infected roots by a series of centrifugations in 50% sucrose followed by a continuous gradient of Ludox^®^ (Ludox^®^ HS 40, a colloidal silica 40 wt. % suspension in H_2_O, density 1.3 g mL^−1^ at room temperature) to the separate spores from other contaminants. Xu et al. [18] developed a much more practical method of a Ludox^®^ density gradient centrifugation for extracting ciliates in marine sediments. This method has a higher efficiency whilst minimizing the number of centrifugation steps.

In recent years, microbial biology and host–pathogen interactions have benefited considerably from technological advances in the global identification and quantification of proteins via mass spectrometry (MS)-based shotgun approaches. Huang et al. [19] identified 18 spore-specific proteins in *Cryptococcus neoformans* using proteomic profiling between spores and vegetative cells. In another example, a complementary analysis of the conidial surface proteome in *Aspergillus fumigatus* identified 148 proteins which are essential for the virulence of the pathogen [20]. Spore proteomic studies have been carried out using gel-based techniques [21], isobaric labelling [22] and label-free shotgun proteomics [23]. Isotope labelling and gel-based approaches may involve several technical difficulties, such as the specific requirements for metabolic labelling, while label-free methods are straightforward and correlate well with isotope label-based quantification [24]. In soil-borne pathogens with strong cell walls, both cell lysis and protein recovery remain the most difficult steps in the proteomic studies of these organisms [25].

Many efficient sample preparation methods are available for the MS-based proteomic analysis of microorganisms when samples are unlimited. Recent innovations include the SP3 (single-pot solid-phase-enhanced sample preparation method) [26] and the suspension-trapping filter-based approach (S-Trap) developed by Zougman et al. [27]. The SP3 protocol consists of nonselective protein binding and uses a hydrophilic interaction mechanism for the removal of non-protein components. In this method, proteins are captured on the surface of magnetic beads which facilitates the following washing steps and the recovery of peptides [26]. For the S-Trap method, the proteins are trapped in a filter and the contaminants such as detergents, chaotropic agents, salts, buffers, acids, and solvents are removed in a short wash step [28]. Hayoun et al. [29] compared the sample preparation methods including in-gel proteolysis, SP3, and S-Trap methods for the MS analysis of microorganisms. Their study demonstrated that SP3 delivers a higher coverage of the sample with greater numbers of identified proteins. However, Doellinger et al. [30] showed that in the presence of sodium dodecyl sulphate (SDS), the SP3 method identified the lowest number of peptides, which highlights a potential limitation of this technique where high concentrations of detergent are required.

Due to the difficulties with sporosori isolation and purification, there have been no reports on the practical application of MS-based shotgun proteomics for the determination of the mechanism underlying the specific biological processes during the formation and germination of resting spores in *S. subterranea.* In the present study, we first tested the efficiency of the Ludox^®^ density gradient centrifugation, originally designed for marine sediments, for the purification of *S. subterranea* resting spores from potato tissue and other contaminants. Then, we examined the quantitative differences in proteins from *S. subterranea* that were prepared under various conditions. We compared the performances of a urea lysis buffer followed by the SP3 technique and an SDS lysis buffer followed by the S-Trap filter for the protein preparation of *S. subterranea*.

## 2. Results and Discussion

*S. subterranea sporosori*, the aggregates of resting spores present in the soil and within the lesions of diseased potato tubers, are critical for pathogen survival, dispersal and for the initiation of new host infections. Host infection follows the stimulation of resting spore germination to release motile zoospores [31]. Understanding the molecular basis of sporosori dormancy and germination are therefore valuable for the development of novel disease interventions.

### 2.1. Sporosori Purification

*S. subterranea* cannot be grown in pure culture on an artificial medium and thus sources of the pathogen will invariably be contaminated with remnant potato tissues and various soil microorganisms [32]. This poses major difficulties for obtaining the pure protein, RNA, or DNA of the pathogen. Here, we utilized a density gradient method using Ludox to separate *S. subterranea* sporosori from the contaminants. After centrifugation in the Ludox^®^ gradient, a band of living organisms including *S. subterranea* sporosori was consistently found one-fourth of the way from the top of the tube (Figure 1a). A viability assay results from purified spores prepared by the Ludox^®^ method showed that purification by the Ludox^®^ density gradient centrifugation did not affect the viability of *S. subterranea* (Figure 1b), with no statistically significant difference. The germination of spores in Hoagland’s solution for both purified and non-purified sporosori was 100% after 6 days, confirming no effect of the treatment on the viability [31].

### 2.2. Optimisation of Protein Preparation for Mass Spectrometry

In order to optimize a protocol for *S. subterranea* MS-based proteomics, two sample preparation methods were compared. Sporosori were first extracted and enriched using Ludox^®^ gradient centrifugation. Protein samples solubilized in either SDS or urea were then prepared for digestion using the S-Trap or SP3 approaches, respectively. According to the numbers of matched peptides and proteins identified, the best performing method was the S-Trap, with an average of 654 proteins and 6658 peptides, whereas the SP3 method resulted in only 189 proteins identified on the basis of 602 matching peptides (Figure 2a,b). After the exclusion of proteins identified on the basis of a single matching peptide, the proteins identified using the S-Trap and SP3 methods were reduced to 595 and 95, respectively (Figure 2c,d).

To evaluate the reproducibility of protein preparation techniques, the number of missing values for each identified protein was calculated. In the S-Trap samples, more than 90% of the identified proteins had no missing intensity values and only ~ 8% of the samples showed one or two missing values in their abundance. On the other hand, in the SP3 samples, 37% and 12% of the proteins presented with two and one missing values, respectively (Figure 3).

Efficient sample preparation for the MS-based analysis often requires the solubilization of cellular and hydrophobic proteins in the presence of either SDS or strong chaotropic reagents. However, such conditions are incompatible with both enzymatic digestion and LC−MS. Among the strategies developed for the sample clean-up, the suspension trapping was found to be superior to both in-solution digestion and other filter-based sample preparation methods [28,33]. The S-Trap is a simple device that traps proteins in a filter and enables the efficient removal of contaminants such as chaotropic agents and detergents prior to digestion. In comparison to the SP3 approach, we found that the S-Trap enabled more efficient digestion leading to higher proteomic coverage in a shorter period and showed an improved reproducibility (Figure 2 and Figure 3). The proteomic analysis of the soil-borne necrotrophic pathogen *Rhizoctonia solani* using in-gel digestion and MALDI-TOF MS identified 130 proteins related to the maturation of fungal sclerotia [34]. In *Colletotrichum acutatum*, 365 proteins were identified using two-dimensional electrophoresis combined with MALDI-TOF/TOF mass spectrometry during the germination of the pathogen [35]. Mappa et al. [36] assessed the ratio of *Bacillus atrophaeus* spores and vegetative cells by shotgun proteomics and identified 602 proteins. The number of proteins identified in other proteomic studies of the spores included 319 in *Moniliophthora perniciosa*, 118 in *Botrytis cinerea* spores and 148 conidial surface proteins in *Aspergillus fumigatus* [20,37,38]. Thus, the present study has identified a comparable number of proteins from the resting spores of *S. subterranea*.

### 2.3. Quantitative Efficiency of Ludox^®^ Purification for Protein Analysis of Resting Spores

To evaluate the efficiency of the spore purification method, we used the S-Trap method for sample preparation, followed by the label-free shotgun proteomic analysis of purified and non-purified sporosori. The quantitative results demonstrated that the number of identified proteins in *S. subterranea* resting spores increased by 38.9% after the Ludox^®^ purification, where 470 and 653 proteins were identified in non-purified and purified resting spores, respectively (Figure 4a). As shown in Figure 4b, although 38 proteins were not detected after the purification of the resting spores, 221 new proteins were identified in the purified spores compared to the non-purified ones. Molecular and functional studies of the powdery scab pathosystem were hampered by the obligate biotrophic nature of the pathogen. *S. subterranea* is not culturable and can only be found on its living host or in the soil as a form of resting spore. Many protocols have been developed for the soil-borne pathogen purification to increase the quality of the extracted DNA, RNA, and protein [16,17,39]. The gradient density centrifugation by Percoll^®^ and Ludox^®^ are the two commonly used methods for the purification of living microorganisms from the contaminants. In contrast to Percoll^®^, Ludox^®^ is cheaper, denser and needs fewer preparation steps [18,40]. In this study, we adapted the Ludox^®^ method for the purification of *S. subterranea* resting spores and characterized any improvement in the resting spore proteome analysis. The results presented here showed that the purification of sporosori increased the quantitative efficiency of the protein analysis of *S. subterranea*.

### 2.4. Major Metabolic Pathways and Cellular Processes of S. subterranea Resting Spores Revealed by a Bioinformatics Approach

Label-free quantitative proteomic LC-MS analyses revealed 653 unique proteins in the *S. subterranea* resting spore’s proteome. These proteins were identified in at least three replicates with a protein false discovery rate of less than 1%. Here, a web-based functional annotation analysis of S. subterranea resting spores’ proteins was used to identify the gene ontology (GO) terms. The identified proteins were classified into three GO categories: molecular functions, cellular components, and biological processes (Figure 5a–c). The largest categories of molecular functions were proteins with catalytic activity (*n* = 243) and binding proteins (*n* = 224), respectively (Figure 5a). In the cellular component category (Figure 5b), 68% of the identified proteins were involved in cellular anatomical entity, 27% in protein-containing complex, and 5% in the cell structure. In the biological process category, the identified proteins were involved in cellular processes (*n* = 209), metabolic processes (*n* = 197), localization (*n* = 43), biological regulation (*n* = 30), cellular component organization (*n* = 12), response to stimulus (*n* = 5), and formaldehyde catabolic processes (*n* = 1) (Figure 5c). The expression analysis of the protein pathway of *S. subterranea* resting spores revealed that the identified proteins were mainly concentrated on the carbohydrate degradation, amino acid biosynthesis, amino acid degradation, and carbohydrate metabolism (Figure 5d). Considering the minimal metabolic activity of resting spores, most of the proteins required for the maintenance of dormancy and the germination of dormant spores must be supplied during sporulation [41,42]. In line with our results here (Figure 5), previous proteomic analyses of resting spores have identified the proteins primarily associated with pyrophosphatase, hydrolase, transferase, and oxidoreductase activities [43], protein folding and degradation [44], the biological processes of protein synthesis, protein metabolism, and energy production [45].

## 3. Materials and Methods

### 3.1. Pathogen Source

*S. subterranea* sporosori were scraped from powdery scab diseased potato tubers collected from commercial fields in Devonport, Tasmania, Australia. The diseased tubers were washed with running tap water for 2 min, rinsed in sterile water, and air dried for 2–3 days. Avoiding excessive potato tuber tissue, all the scab lesions were individually excised using a scalpel, and they were pooled and dried at 40 °C for 3 days and stored at 4 °C until use. Between 50 and 80 tubers were excised to produce 1 g of dried material.

### 3.2. Ludox^®^ Density Gradient Centrifugation for Sporosori Purification

We adapted the method of Xu et al. [18] for sporosori purification. In this protocol, the purification of resting spores could be performed in four steps: 

Dried *S. subterranea* sporosori preparation (100 mg) were macerated in 3 mL sterile water using a mortar and pestle and filtered through two layers of cheesecloth.

The filtrate (2–3 mL) was layered onto 9 mL of Ludox^®^ (HS-40 colloidal silica, Sigma, Macquarie Park, NSW, Australia) in a 15 mL centrifuge tube and with 2 mL distilled water added on the top.

The mixture was centrifuged at 4200× *g* for 15 min at room temperature. Following centrifugation, two bands were recognizable; with *S. subterranea* sporosori found in the uppermost band approximately one quarter of the way from the top of the tube. The lower band contained contaminants.

To remove any remaining Ludox^®^, 2–4 mL of purified extract was transferred to a new 50 mL tube and diluted with 40 mL of sterile water before centrifugation at 4200× *g* for 8 min at room temperature. The supernatant was discarded and the pellet stored at 4 °C until use.

The purity of the *S. subterranea* sporosori preparation was determined by the dilution of a 10 µL aliquot of the purified preparation in distilled water and observation with light microscopy (200×).

### 3.3. Spore Viability

Resting spore viability was assessed by the determination of capacity to germinate and release active zoospores [31]. Sporosori preparations (5 mg each of purified and non-purified), were added to individual 2 mL microcentrifuge tubes, suspended in 1 mL of Hoagland’s solution [44] and incubated at 25 °C in the dark. Ten tubes were used for each treatment. Sub-samples (10 µL) were taken from each tube daily for 7 days and observed microscopically (200–400×) for the presence of active zoospores. The time to the first observation of zoospores was recorded.

### 3.4. Protein Extraction and Digestion

An overview of the optimisation of a protein preparation protocol for *S. subterranea* proteomics is given in Figure 6. The purified sporosori (from 100 mg dried lesion material) were resuspended in lysis buffer (5% SDS, 50 mM ammonium bicarbonate, and protease inhibitor (1 tablet of cOmplete Mini EDTA-free; Roche Diagnostics, North Ryde, NSW, Australia)). The samples were homogenised using a Fast Prep-24 bead beater (4000× *g* for 60 s) using PowerBead tubes, ceramic 2.8 mm (Qiagen, Hilden, Germany) followed by 30 s resting at room temperature. The homogenisation procedure was repeated three times for each sample. The lysates were then clarified by centrifugation at 16,000× *g* for 10 min. The supernatant was collected, 6 volumes of ice-cold acetone were added and the tubes were incubated at −20 °C overnight to precipitate the proteins. Following incubation, the tubes were centrifuged (16,000× *g* for 10 min) and the supernatant was discarded. The pellets were washed three times with chilled acetone, left to air dry for 5–10 min at room temperature and resuspended in either urea buffer (7 M urea and 2 M thiourea in 40 mM Tris, pH 8.0) or SDS buffer (5% SDS and 50 mM ammonium bicarbonate) for the next step.

#### 3.4.1. Single-Pot Solid-Phase-Enhanced Sample Preparation (SP3 Method)

The protein samples were quantified using the Qubit protein assay (Thermo Fisher Scientific, Waltham, MA, USA) and diluted to approximately 1 mg/mL in denaturation buffer (7 M urea and 2 M thiourea in 40 mM Tris, pH 8.0). Aliquots of 30 µg protein were sequentially reduced using 10 mM DTT overnight at 4 °C, alkylated using 50 mM iodoacetamide for 2 hrs at room temperature and then digested with 1.2 μg proteomics-grade trypsin/LysC (Promega, Madison, WI, USA) according to the published SP3 protocol [26]. The digests were acidified by the addition of trifluoroacetic acid to 0.1% and the peptides were collected by centrifugation at 21,000× *g* for 20 min. The samples were further cleaned up by offline desalting using ZipTips (Merck, Darmstadt, Germany) according to the manufacturer’s instructions.

#### 3.4.2. Suspension-Trapping (S-Trap Method)

Proteins in the SDS buffer were reduced by adding 20 mM dithiothreitol (10 min at 95 °C) followed by alkylation with 40 mM iodoacetamide (30 min at room temperature in the dark). The samples were prepared according to the S-Trap microcolumns (Protifi, Farmingdale, NY, USA) manufacturer’s instructions. In brief, the samples were acidified with a final concentration of 1.2% phosphoric acid. To aggregate the proteins in colloidal particles, the samples were diluted with 6 volumes of S-Trap protein binding buffer (100 mM ammonium bicarbonate in 90 % aqueous methanol, pH 7.1) and each sample was loaded onto an S-Trap micro spin column. The proteins were trapped in the filter by centrifugations at 2800× *g* for 1 min and washed 3 times with 150 μL of S-Trap buffer. Finally, 2 μg of sequencing-grade trypsin in 20 µL of 50 mM ammonium bicarbonate were added into the filter and digested at 47 °C for 1 h. The peptides were eluted using 50 mM ammonium bicarbonate and then 0.2% formic acid and vacuum centrifuged to dry. The samples were further cleaned up by offline desalting using ZipTips (Merck, Darmstadt, Germany) according to the manufacturer’s instructions.

### 3.5. LC–MS/MS Analysis

The peptide samples equivalent to 1 mg were separated using an Ultimate 3000 nano RSLC system (Thermo Fisher Scientific, MA, USA). The peptides were first concentrated on a 20 mm × 75 μm PepMap 100 trapping column (3 μm C18) for 5 minutes, then separated using a 250 mm × 75 μm PepMap 100 RSLC column (2 μm C18) at a flow rate of 300 nL/min and held at 45 °C. A 90-min gradient from a 98% mobile phase A (0.1% formic acid in water) to 50% mobile phase B (0.08% formic acid in 80% acetonitrile and 20% water) comprised the following steps: 2–10% B over 12 min, 10–25% B over 48 min, 25–45% B over 10 min, holding at 95% B for 5 min then re-equilibration in 2% B for 15 min. The nano HPLC system was coupled to a Q-Exactive HF mass spectrometer equipped with a nanospray Flex ion source (Thermo Fisher Scientific, Waltham, MA, USA) and controlled using Xcalibur 4.1 software. The spray voltage was set to 2.0 kV, S-lens RF level to 50, and a heated capillary set at 250 °C. The MS scans were acquired from 370–1500 *m*/*z* at a 60,000 resolution, with an AGC target of 3 × 10^6^ and a maximum injection time of 100 ms. The fragment ion scans were acquired at a 15,000 resolution (scan range 200–2000 *m*/*z*), with an AGC target of 2 × 10^5^ and a maximum injection time of 28 ms. An isolation width of 1.4 *m*/*z* was used, and the normalized collision energy for HCD was set to 27. The MS/MS spectra were acquired in a data-dependent mode using a Top15 method with 30 s dynamic exclusion of fragmented peptides.

### 3.6. Data Processing

The MS/MS raw data were investigated against the *S. subterranea* database in UniProt (https://www.uniprot.org/proteomes/, containing 11,129 proteins) using the Andromeda search engine in MaxQuant software (v. 1.6.0.16, www.maxquant.org) with default search settings for Obitrap MS and the match-between-runs function enabled only for the proteomic analysis of purified and non-purified sporosori. The proteins were identified based on the label-free quantification (LFQ) values reported by MaxQuant and based on at least two unique peptides. A filter was then applied to include only those proteins detected in a minimum of three replicates. The proteins labelled as either reverse hits or contaminants were removed from the final analysis. Based on a normal distribution of protein abundances, the missing values were replaced with random intensity values. The clustering correlation analyses were performed in the Perseus software (v. 1.5.0.15, www.perseus-framework.org). All the data were exported from Perseus into Excel after each data-processing step (Appendix A). Four biological replicates were used for the protein extraction from purified and non-purified samples, while three technical replicates were used for the peptide sample preparation.

## 4. Conclusions

This paper marks the first report of method development for the purification and proteomic analysis of the resting spores of the obligate plant pathogen *S. subterranea*, a process confounded by the aggregation of resting spores into complex sporosori structures of unequal size. A deeper understanding of the molecular basis of resting spore germination will be critical for the development of novel approaches for the management of persistent soil inoculum [45]. For proteomic studies, we showed the benefits of Ludox^®^ density gradient centrifugation purification of *S. subterranea* sporosori from a complex sample that may contain very diverse contaminants with an increase the numbers of identified proteins of approximately 40%. The Ludox^®^ purification protocol provides a cleaner suspension of the resting spores which should be useful for the protein analysis and comparative molecular studies of *S. subterranea*. These procedures remove most of the contaminating debris and microorganisms without affecting the viability of spores. The SP3 method and on-filter proteolysis of extracted proteins were optimized to improve the label-free quantitative proteomics of *S. subterranea*. In comparison with the SP3 method, the S-Trap method delivered a higher number of protein identifications with an improved reproducibility and will form the foundation of our further analysis of resting spore germination and disease development in *S. subterranea*. Overall, these data provide optimised procedures for the global proteome analyses of the obligate biotroph *S. subterranea* and we anticipate that it may be applicable for other similarly difficult non-culturable pathogens.

## Figures and Tables

**Figure 1 molecules-25-03109-f001:**
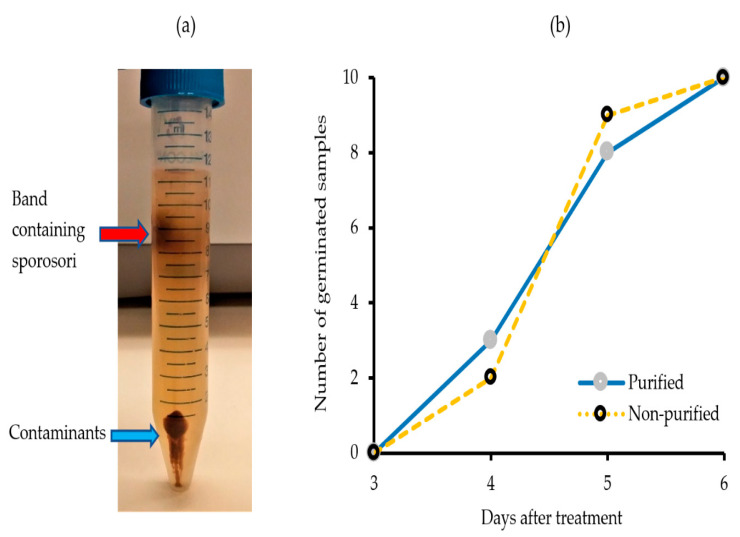
(**a**) Band of resting spores in the Ludox^®^ gradient and (**b**) the cumulative curve of purified and non-purified *S. subterranea* resting on the spore’s germination in the Hoagland’s solution.

**Figure 2 molecules-25-03109-f002:**
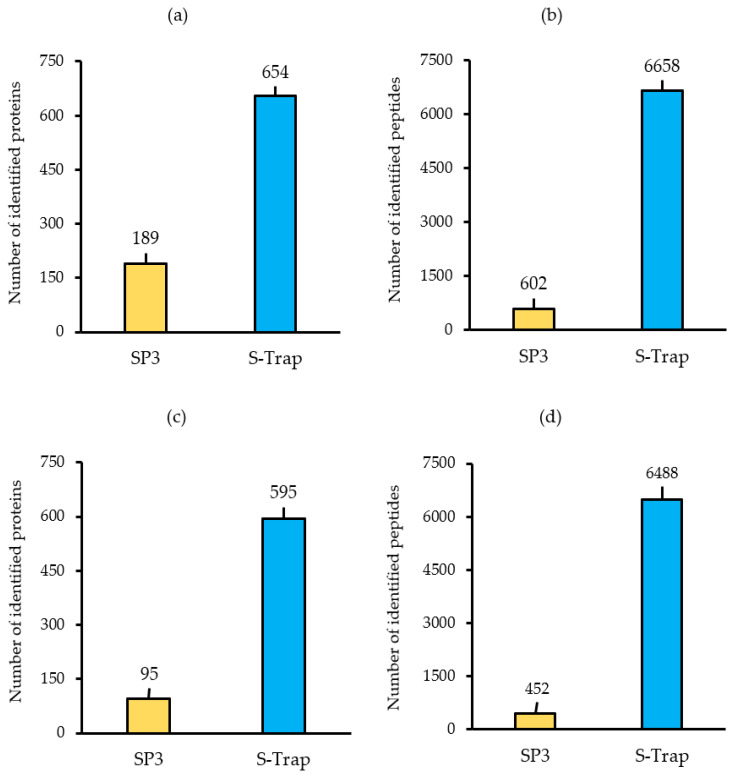
(**a**) Total number of proteins and (**b**) the total number of peptides identified in the database searches. Total proteins were filtered based on at least two unique peptides and include only those proteins detected in a minimum of three replicates (**c**,**d**). SP3: single-pot solid-phase-enhanced sample preparation, S-Trap: suspension-trapping.

**Figure 3 molecules-25-03109-f003:**
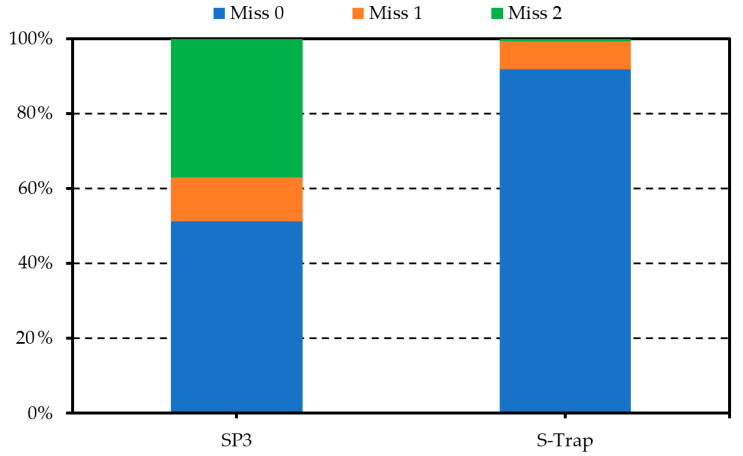
The percentage of proteins with no missing values (Miss 0), one missing value (Miss 1), and two missing values (Miss 2).

**Figure 4 molecules-25-03109-f004:**
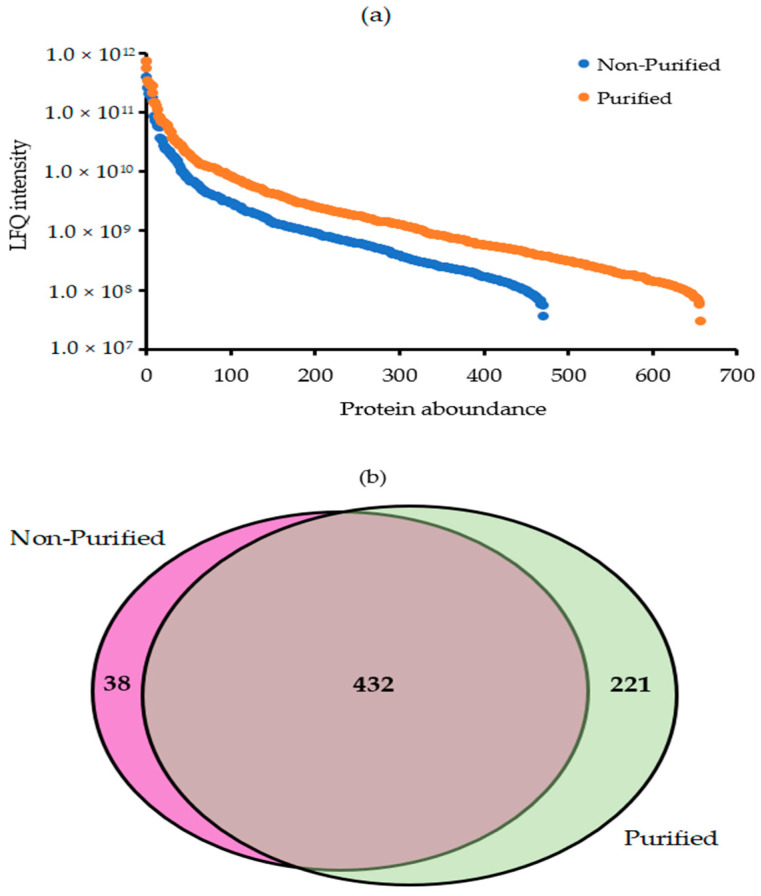
(**a**) The ranked protein abundance plots for the purified and non-purified material; (**b**) the Venn diagram summarizes the groupings of the identified proteins in the *S. subterranea* proteome database. LFQ: label-free quantification.

**Figure 5 molecules-25-03109-f005:**
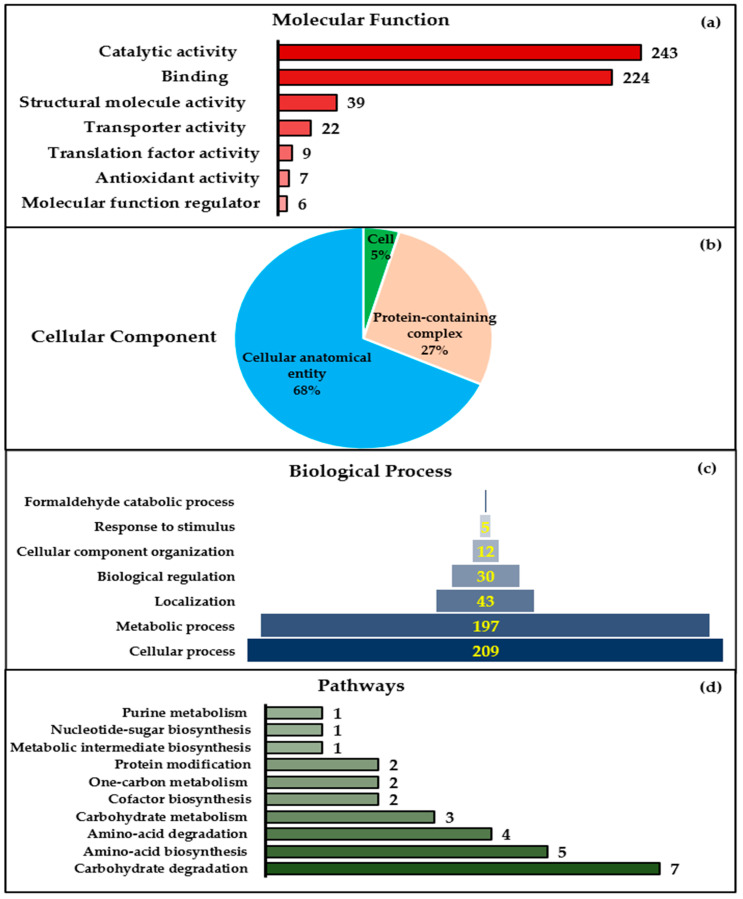
Functional annotation analysis of the *S. subterranea* resting spores’ proteins (FDR ≤ 5%) based on gene ontology (GO) terms (**a**) molecular function, (**b**) cellular components, (**c**) biological processes and (**d**) pathways.

**Figure 6 molecules-25-03109-f006:**
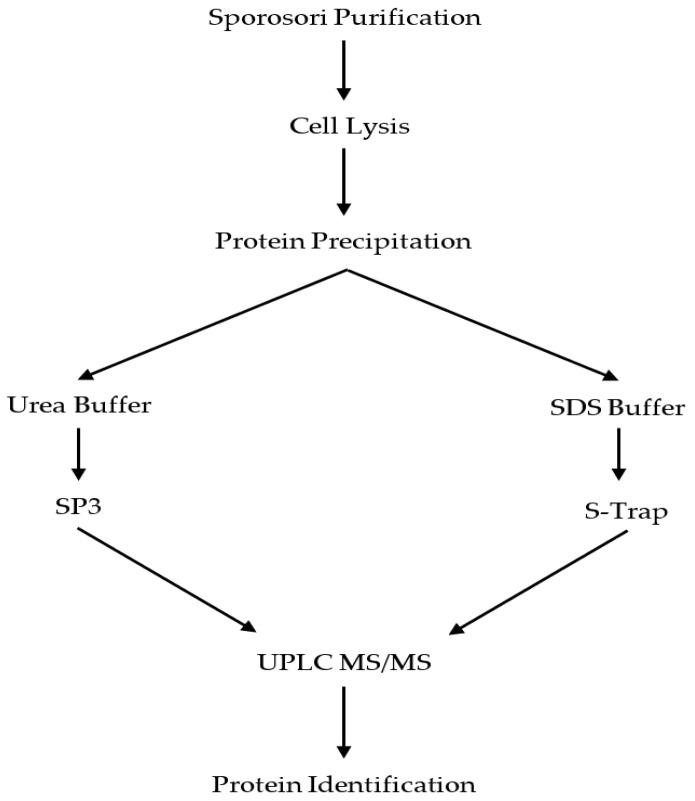
Workflow for the sample preparation optimisation experiments. Proteins from *S. subterranea* sporosori were prepared by the SP3 method or the S-Trap filters and analysed by an MS-based approach. SDS: sodium dodecyl sulphate, UPLC: ultra-performance liquid chromatography.

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
