# Peer review of "Optimisation of Sporosori Purification and Protein Extraction Techniques for the Biotrophic Protozoan Plant Pathogen Spongospora subterranea"

_molecules, 2020, doi:10.3390/molecules25143109_

Round 1
Reviewer 1 Report
The manuscript entitled “Optimisation of sporosori purification and protein extraction techniques for the biotrophic protozoan plant pathogen Spongospora subterranea” reported the optimized method for the purification of sporosori of Spongospora subterranea and subsequently protein extraction. This report provided an effective research technique for powdery scab diseases study in potato. Some revisions are needed as following:
Please check the manuscript carefully, including spell and grammar.
Introduction; Discussion
Line 38 “Octomyxa” should be in italic.
Line 98 “chaotropic, salts, buffers,”? should be corrected to “chaotropic salts buffers,”?
The novel points of the methods and the advantages should be emphasized in the sections of introduction and discussion.
Materials and Methods:
Line 231 “gr” should be better changed to “g”.
Line 272 the capital and small letter of experiment reagent names, such as “Ammonium bicarbonate” or “ammonium bicarbonate”, should be kept consistent. Please check through the text.
Line 343 and 349 please add the versions of software into the brackets.
Author Response
We thank the reviewer for their comments and suggestions
we have accepted all suggestions and have made each appropriate changes in the revised manuscript
Reviewer 2 Report
Reviewer’s comment to the author
Molecules
Dear author,
I have thoroughly reviewed this manuscript titled “Optimisation of sporosori purification and protein extraction techniques for the biotrophic protozoan plant pathogen Spongospora subterranea”.
The authors have done good work. The content of the research manuscript is exciting, timely topic, and its output will be of immense benefit in the field of protein preparation methods for proteomics study of obligate biotrophic pathogens.
Although the manuscript is worthy of being published in this high esteem journal due to the vital information embedded in it.
Overall, I recommend this article as an “accepted” in the present form with some typographical mistakes.
Author Response
We thank the reviewer for their reading of our manuscript and kind comments.
In our revision was have identified a few typographical errors which have been addressed.